# Greater effects of mutual cooperation and defection on subsequent cooperation in direct reciprocity games than generalized reciprocity games: Behavioral experiments and analysis using multilevel models

**Yutaka Horita** [ORCID] *

Department of Psychology, Teikyo University, Tokyo, Japan

* horita@main.teikyo-u.ac.jp

**Data Availability Statement:** Data, code for analysis, and instructions for the experiments

## Abstract

Reciprocity toward a partner's cooperation is a fundamental behavioral strategy underlying human cooperation not only in interactions with familiar persons but also with strangers. However, a strategy that takes into account not only one's partner's previous action but also one's own previous action—such as a win-stay lose-shift strategy or variants of reinforcement learning—has also been considered an advantageous strategy. This study investigated empirically how behavioral models can be used to explain the variances in cooperative behavior among people. To do this, we considered games involving either direct reciprocity (an iterated prisoner's dilemma) or generalized reciprocity (a gift-giving game). Multilevel models incorporating inter-individual behavioral differences were fitted to experimental data using Bayesian inference. The results indicate that for these two types of games, a model that considers both one's own and one's partner's previous actions fits the empirical data better than the other models. In the direct reciprocity game, mutual cooperation or defection—rather than relying solely on one's partner's previous actions—affected the increase or decrease, respectively, in subsequent cooperation. Whereas in the generalized reciprocity game, a weaker effect of mutual cooperation or defection on subsequent cooperation was observed.

## Introduction

Humans cooperate with other people, even with strangers and non-relatives, to establish a large-scale society. Cooperation is defined as a type of behavior that implies sacrificing personal interests and thereby providing benefits to others. In evolutionary game theory, the prisoner's dilemma (PD) is used as a standard model to examine the evolution of cooperation. In PD, two players can decide either to cooperate (C) or to defect (D). If both players mutually cooperate, they each receive a reward $R$. If they mutually defect, they each receive a punishment $P$. If one decides to cooperate and the other one to defect (CD), the cooperator receives

(translated into English) are available at the Open Science Framework: https://osf.io/5aqkh/.

**Funding:** YH was supported by JSPS KAKENHI (Grant No. JP18K13276).

**Competing interests:** The authors have declared that no competing interests exist.

the payoff $S$ and the other the payoff $T$. The payoff structure of the PD is given by the following equation: $T > R > P > S$ (and $2R > T + S$). Without employing any additional mechanisms, natural selection considers defection as more strategic. This is because cooperation is often costly for individuals; on the contrary, defection means yielding immediate benefits for them. However, evolutionary game theory defines several conditions under which the cooperative strategy can compete with the non-cooperative one [1–5]. Specifically, reciprocity—a behavioral rule that depends on a previous action of a counterpart—is considered a key concept underlying the evolution of cooperation between humans or animals.

Repeated interaction with the same opponent is one known mechanism for facilitating cooperation, and it is referred to as "direct reciprocity" or "reciprocal altruism" [6–8]. If the probability of repeating an interaction between the same individuals is high, a reciprocal strategy can be considered as a payoff-maximizing one. In a repeated and simultaneous PD, tit-for-tat (TFT) is a well-known successful strategy that implies copying the previous action of an opponent [6, 7]. TFT can be applied to establish mutual cooperation through cooperative strategies and to avoid being exploited by unconditional defectors.

The win-stay lose-shift (WSLS) strategy—also referred to as the Pavlov strategy—is one of the most successful strategies used in the simultaneous PD [9]. WSLS is a variant of reinforcement learning, which suggests repeating the previous action that yields higher payoffs to the focal player and changing his/her behavior if he/she obtains lower payoffs. In the PD, WSLS is a strategy that incorporates not only the opponent's but also one's own previous action; it suggests cooperating after mutual cooperation (CC) or mutual defection (DD) and defecting after exploitation (DC) or being exploited (CD). Theoretically, WSLS can outperform TFT in an iterated and simultaneous PD under the condition that errors can occur [9]. For instance, a player may misperceive that his/her opponent has defected even though the opponent has in fact cooperated. In such a situation, TFT can easily fall into mutual defection. Moreover, WSLS can outperform TFT, which is attributed to its several advantages. First, WSLS enables the correction of errors; for example, after DD, it can switch to cooperation, whereas TFT implies repeating defection. Second, unlike TFT, WSLS can exploit unconditional cooperators.

Although direct reciprocity can explain how a cooperative strategy emerges during repeated interactions between two persons, reciprocal cooperation beyond an iterated relationship also is widely observed in human society. Such reciprocal behavior can be described by the following simple rule: "If I receive help from my partner, I will help the other person." Such a form of reciprocity is referred to as "generalized reciprocity" (or "upstream reciprocity"). Some empirical studies have shown that reciprocal cooperation can occur even with generalized reciprocity [10–17]. However, other empirical studies have suggested that cooperation based on generalized reciprocity is unstable and weak [17, 18]. Theoretical research has demonstrated that cooperation based on generalized reciprocity can be established under strict conditions, such as a small population size [19, 20]. It also suggests that a strategy without cognitive complexity—such as a WSLS-like one—can perform better and that cooperation can be sustained in the case of generalized reciprocity [20, 21].

In addition to theoretical works, empirical studies have emphasized the important role of reciprocity toward the other's cooperation [22, 23]. On the other hand, recent empirical studies have indicated the role of other behavioral models for predicting human behavior in experiments. Models that focus on one's own payoffs—such as reinforcement learning—can explain human behavioral patterns appropriately in social dilemma experiments in which many individuals decide whether or not to cooperate for their group [24–26]. A behavioral rule that considers not only the counterpart's previous action but also one's own has been observed in several social dilemma cases [26–31]. In the iterated PD game, the particular proportion of participants who employ a WSLS-like strategy has been indicated [32, 33]. In the generalized

reciprocity situation, some empirical studies have suggested that people behave in a reciprocal manner [12–16]. To the best of our knowledge, however, it is still unclear whether or not the WSLS-like strategy plays an important role in explaining real human behavior in experimental situations of generalized reciprocity.

In this study, we aim to investigate the possibility of constructing a comprehensive model capable of encompassing the variety of individuals' cooperative behaviors in interactions with a given person or with strangers. We conducted two types of experimental games: direct and generalized reciprocity games. Each participant can decide whether to donate (cooperate) or not to donate (defect) money repeatedly to the same person in the direct reciprocity game and to different persons in the generalized reciprocity game, respectively. We fitted several models to predict the probability of cooperation and compared the goodness-of-fit estimates for each model.

We compared the predictive accuracy of each model using a model comparison approach with widely applicable information criteria (WAIC) [34]. Complex models that include many parameters fit the data better than simple models. However, the complex models have a trade-off between overfitting the observed data and hurting predictive accuracy. A model that uses both ones' own and one's partner's action as predictor variables can describe both TFT-like and WSLS-like behavior. However, if the strategy that implies simple reciprocity toward the partner's previous action (e.g., TFT) is sufficient to explain the various patterns of human cooperation, parameters that depend on one's own action are redundant. According to previous studies [26–31], we examined whether a strategy that depends on the combination of one's own and one's partner's previous actions can explain the experimental data better even in direct and generalized reciprocal situations.

As discussed above, various strategies of cooperation have been proposed, such as TFT and WSLS (or reinforcement learning). However, as shown in a series of previous experimental studies [22, 35–37], the tendency for cooperation differs among individuals. Therefore, we fitted multilevel models that considered the variance in behavior among individuals to the empirical data through the method of Bayesian analysis, including Markov chain Monte Carlo (MCMC) simulations. A multilevel model assumes that the parameter values vary among individuals and are drawn from a group-level population distribution. Bayesian inference with multilevel models can estimate "posterior distributions" of parameters (namely, intervals of parameters) both for a group-level population and for each individual. Bayesian MCMC simulations can be used to fit complicated multilevel models that include multiple varying effects to the experimental data. Considering inter-individual differences in cooperative tendency, we investigate whether or not one's own and one's partner's previous behaviors are important in explaining the experimental data.

It was expected that the behavioral patterns in the generalized reciprocity game would differ from those in the direct reciprocity game. Costly cooperation would be rewarded by the opponent's immediate cooperation in the direct reciprocity game, and it would increase one's future profits, yielded by the achievement of mutual cooperation. However, the immediate return of cooperation was not expected in the generalized reciprocity game. Therefore, how important the players consider achieving mutual cooperation as a goal would differ between the two games. However, as described above, theoretical models considering one's own and other's actions, such as the WSLS-like strategy, have been proposed to explain the cooperation in generalized reciprocity [20, 21]. To investigate whether or not different behavioral patterns depending on one's own and one's partner's actions would be observed between two types of dyadic interaction (i.e., interaction with the same individual or interaction with strangers), we used a model that considered one's own and one's partner's previous actions as a candidate model for both the direct and generalized reciprocity games.

## Materials and methods

### Experiments

In this study, 40 undergraduate students (21 women and 19 men with an average age of 20.00 and a standard deviation of 1.66) were invited to participate in the direct reciprocity game, while 40 other undergraduate students (17 women and 23 men with an average age of 19.95 and a standard deviation of 1.48) were involved in the generalized reciprocity game. Participants were recruited from a participant pool via e-mail, and monetary rewards were provided for their participation.

The current study was approved by the ethic committee of Teikyo University, Tokyo, Japan. Participants signed an informed consent form before participating in an experiment.

**Direct reciprocity game.** Six or four participants played an iterated PD game in each single experimental session, with eight experimental sessions conducted in total. Each participant in a single session was paired with one of the other participants. Both participants in a pair received 20 yen (approximately 18 US cents) from the experimenter as an endowment and could decide either to give the money to his/her partner (cooperate) or to retain it (defect). If the participant gave the money to his/her partner, he/she lost the money, whereas the partner received the doubled sum of the money (40 yen). However, if the participant did not give away the endowment, he/she kept it, and the partner received nothing. The pair of participants was not changed until the end of the game (Fig 1A). Before making each decision, participants were reminded whether or not their partner had given money to them in the previous decision (S1 Fig). They submitted the decision to the partner repeatedly 42 times in total.

**Generalized reciprocity game.** Five participants played a gift-giving game in each single experimental session, and eight experimental sessions were conducted in total. Each participant was paired with one of the other participants. One participant in the pair was assigned the role of either a donor or a recipient. The donor received 20 yen from the experimenter as an endowment and could decide either to give the money to the recipient (cooperate) or to retain it (defect). When the donor gave away the money, he/she lost the money, whereas the recipient received the doubled sum of money (40 yen). When the donor did not give away the

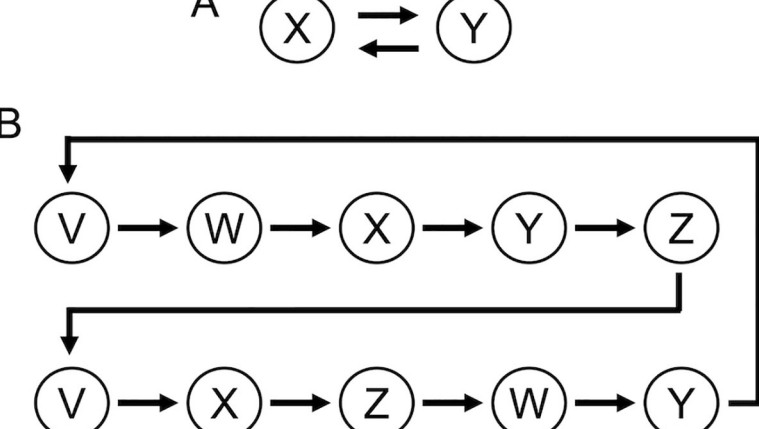

**Fig 1. Examples of each game.** (A) Direct reciprocity game. Two players (player X and Y) are paired and each player repeatedly decides whether or not to cooperate with his/her partner. After each decision, each player is informed whether or not his/her partner has cooperated. (B) Generalized reciprocity game. In a group of five players, each player subsequently decides whether or not to cooperate with his/her neighbor. Players V, W, X, Y, and Z submit their decisions to his/her downstream neighbor in this order. After each decision, each player is informed whether or not his/her upstream neighbor had cooperated.

endowment, he/she kept the money, and the recipient received nothing. After the donor made his/her decision, his/her decision was relayed to the recipient.

In the generalized reciprocity game, all participants contributed to the chain of decisions and submitted his/her decision sequentially, as depicted schematically in Fig 1B. Players V, W, X, Y, and Z submitted the decisions in this order. Each participant submitted his/her decision two times in a single chain of decisions. For instance, player V first made the decision as a donor, and then player W was informed about player V's decision. Next, player W could decide whether or not to give money to player X. In the similar manner, players Y and Z also made their decisions sequentially. After player Y made a decision toward player V, one rotation of the chain was terminated, and five players made decisions again in the same order. Seven rotations were run for each single chain, and three independent chains of decision were run simultaneously (see S2 Fig). Therefore, each participant submitted a decision 42 times (= 2 decisions × 7 rotations × 3 chains) in total.

**Procedure.**   Upon arrival, participants were accompanied to the laboratory where tablet computers were deployed on desks. Each participant sat in front of a tablet computer. While participating in an experimental game or answering questionnaires, partitions were placed between players so that they could not see each other's faces or displays.

After all participants had arrived in the laboratory, the experimenter started explaining the rules of either the direct or the generalized reciprocity games. The experimental procedure was presented using audible slides developed in PowerPoint. Participants were also provided with written instruction sheets so that they could put them on their desks during the experiment. After completing this instruction, participants answered questions to confirm their understanding of the rules of the game. After all participants answered the questions correctly, the game was initiated. The experimental game was conducted using tablet computers connected via a Wi-Fi network. The programs for games were implemented in z-Tree [38]. Samples of decision screens displayed to the participants during the game are presented in S1 Fig.

To assure anonymity, the decisions of each participant were recorded using a randomly assigned number. In addition, each participant was assigned a computer-generated random three-letter pseudonym. When a participant made a decision, the pseudonyms of other participants were displayed on his/her computer screen (S1 Fig). In the direct reciprocity game, the pseudonyms assigned to each participant were never changed until the end of the game. The name of a partner was displayed on the screen as a single pseudonym that remained the same during the course of the game. Conversely, in the generalized reciprocity game, once each participant submitted his/her decision, a new pseudonym was assigned to him/her. In this game, the same pseudonym could not be utilized again during the game, and participants were shown different pseudonyms in every decision round.

When a participant submitted a decision, the previously made decisions of both the participant and his/her partner were displayed on the computer screen (see S1 Fig as an example). The participants were informed about his/her opponent's previous decision in the direct reciprocity game, and about his/her upstream neighbor's decision in the generalized one. Participants were also informed that there is a probability that errors can occur; i.e., the opposite of the partner's actual decision was occasionally displayed to the participant. This procedure was used in order to establish a situation in which WSLS performed better theoretically [9] and to obtain as many observations as possible about the way participants reacted to their partner's cooperation or defection. In case of an error, a participant was likely to be informed that the partner did not give away money, even if the partner did the opposite and vice versa. The probability of an error was calculated at 25%; however, the exact probability value was not revealed to the participants.

After completing the game, each participant was paid individually according to the earnings acquired during the game. On average, the participants who played the direct reciprocity game received 1,340 yen (approximately 12.2 US dollars), whereas those who played the generalized reciprocity game received 1,265 yen. Each experiment took approximately an hour.

## Model fitting

**Multilevel models.** In this research, four models that predict the probability of cooperation were fitted to the experimental data. Here $p$ is the probability of cooperation; $y$ is a binary value that denotes the decision of a participant (0 = defection; 1 = cooperation). Each decision is assumed to obey a Bernoulli distribution with probability $p$:

$$y \sim \text{Bernoulli}(p). \tag{1}$$

The models assume a multilevel structure of parameters, which means that the values of the parameters vary depending on the player. A certain parameter corresponding to player $i$ that affects a response variable (namely, decision of the player) is denoted by $m_i$. The parameter $m_i$ is assumed to be drawn from a normal distribution with a mean of $\mu$ and a standard deviation of $\sigma$:

$$\begin{aligned} m_i &= \mu + z_i \sigma, \\ z_i &\sim \text{Normal}(0, 1), \end{aligned} \tag{2}$$

where $z$ represents a standardized score (z-score) for the parameter for each individual. It is assumed to obey a normal distribution with a mean of 0 and a standard deviation of 1. The quantities $\mu$ and $\sigma$ are hyperparameters that determine the parameters of each player ($m_i$), and $m_i$ is referred to as the varying effect. Here $\mu$ represents a group-level effect considered to predict the response variable. In the multilevel model, each individual's parameter shrinks toward the group-level mean (namely, the hyperparameter). This statistical phenomenon is called "shrinkage," and it prevents each individual parameter from becoming an outlier. The details of estimating these parameters are explained in the S1 Text.

**Partner's action model.** The partner's action (PA) model represents reciprocity toward the partner's previous action. The model can be formulated as follows:

$$\begin{aligned} p &= \frac{\exp(\alpha)}{1 + \exp(\alpha)}, \\ \alpha &= \begin{cases} \alpha_{1,i} + \alpha_{2,i} P_{i,t-1} \\ \nu \ (\text{if } P_{i,t-1} \text{ is a missing value}) \end{cases}, \\ \alpha_{1,i} &= \mu_{\alpha 1} + z_{\alpha 1,i} \sigma_{\alpha 1}, \\ \alpha_{2,i} &= \mu_{\alpha 2} + z_{\alpha 2,i} \sigma_{\alpha 2}, \end{aligned} \tag{3}$$

where $P_{i,t-1}$ denotes the action of player $i$'s partner (namely, an opponent in the direct reciprocity game or an upstream neighbor in the generalized one) in round $t-1$ (1 = cooperation; 0 = defection), which is displayed to player $i$. Note that the previous action of the partner can be presented erroneously in particular cases due to occasional errors. Here, $\alpha_{1,i}$, $\alpha_{2,i}$, and $\nu$ are the parameters and all of them can range from $-\infty$ to $\infty$. The first line of the Eq (3) is called the "inverse logit function", and the function transformed the inferred parameter values into probabilities ranging from 0 to 1. For the following other models in the same manner, the probability of cooperation is estimated using the inverse logit function.

$\alpha_{1,i}$ and $\alpha_{2,i}$ represent the intercept and slope coefficients affecting cooperation, respectively, and $v$ denotes the cooperative tendency when the player is not informed about his/her partner's decision (namely, the decisions at the first round in the direct reciprocity game or the decisions made by the participants assigned as the first elements of a decision chain in the generalized reciprocity game). When we consider the hyperparameter $v$ for the data in the generalized reciprocity game, it could not remove particular divergent transitions, and the efficiency of sampling posterior distributions could deteriorate [39]. Therefore, we used the same parameter value $v$ for all participants. (a normal distribution with a mean of 0 and a standard deviation of 10 was set as a prior for $v$).

**Own and partner's action model.** In the own and partner's action (OPA) model, $p$ is conditioned according to the combination of both the focal player's and his/her partner's previous actions. The model can be described as follows:

$$p = \frac{\exp(\beta)}{1 + \exp(\beta)},$$

$$\beta = \begin{cases} \beta_{1,i} + \beta_{2,i}O_{i,t-1} + \beta_{3,i}P_{i,t-1} + \beta_{4,i}O_{i,t-1}P_{i,t-1} \\ v \text{ (if } O_{i,t-1} \text{ or } P_{i,t-1} \text{ is a missing value)} \end{cases},$$

$$\beta_{1,i} = \mu_{\beta1} + z_{\beta1,i}\sigma_{\beta1},$$
$$\beta_{2,i} = \mu_{\beta2} + Z_{\beta2,i}\sigma_{\beta2},$$
$$\beta_{3,i} = \mu_{\beta3} + z_{\beta3,i}\sigma_{\beta3},$$
$$\beta_{4,i} = \mu_{\beta4} + z_{\beta4,i}\sigma_{\beta4},$$

(4)

where $O_{i,t-1}$ represents player $i$'s action in round $t–1$ (1 = cooperation; 0 = defection); $\beta_{1,i}$, $\beta_{2,i}$, $\beta_{3,i}$, and $\beta_{4,i}$ are parameters based on linear regressions; and $v$ is a parameter of the cooperative tendency in cases when information about previous actions is not provided. As in the PA model, we used the same parameter value for $v$ for all participants. All parameters, $\beta_{1,i}$, $\beta_{2,i}$, $\beta_{3,i}$, $\beta_{4,i}$ and $v$, can range from $-\infty$ to $\infty$.

**Own action model.** For comparison with the above two models, we fitted the own action (OA) model, which assumes that cooperation only depends on the previous action of the focal player. The model was formulated as follows:

$$p = \frac{\exp(\gamma)}{1 + \exp(\gamma)},$$

$$\gamma = \begin{cases} \gamma_{1,i} + \gamma_{2,i}O_{i,t-1} \\ v \text{ (if } O_{i,t-1} \text{ is a missing value)} \end{cases},$$

$$\gamma_{1,i} = \mu_{\gamma1} + z_{\gamma1,i}\sigma_{\gamma1},$$
$$\gamma_{2,i} = \mu_{\gamma2} + z_{\gamma2,i}\sigma_{\gamma2},$$

(5)

where $\gamma_{1,i}$, $\gamma_{2,i}$, and $v$ are the parameters, and all of them can range from $-\infty$ to $\infty$; $\gamma_{1,i}$, and $\gamma_{2,i}$ represent the intercept and slope coefficients affecting cooperation, respectively; and $v$ denotes the cooperative tendency when the focal player makes his/her decision for the first time.

**Null model.**   For another comparison with the PA and OPA models, we fitted the following null model, which includes only a varying intercept for each participant.

$$p = \varepsilon,$$
$$\varepsilon = \frac{\exp(\varepsilon_i)}{1 + \exp(\varepsilon_i)} \quad (6)$$
$$\varepsilon_i = \mu_\varepsilon + z_{\varepsilon,i}\sigma_\varepsilon.$$

In this model, the probability of cooperation by each participant is always determined by a single parameter $\varepsilon_i$, which does not depend on other predictor variables: $\varepsilon_i$ can range from $-\infty$ to $\infty$.

**MCMC simulations.**   For each model, the posterior distributions of the parameter values are inferred through MCMC simulations with four independent Markov chains, conducting a total of 5,000 iterations per chain. First, 2,000 iterations are discarded as warm-up iterations; therefore, 12,000 MCMC samples are utilized in total. Here, $\hat{R}$ values [40] are used to evaluate the convergence of MCMC simulations, and we check whether all parameters in each model converge (that is, whether the $\hat{R}$ values are close to 1.00). The MCMC simulations are implemented using stan and rstan package 2.19.3 provided in R 3.6.3 [41, 42]. Stan utilizes a Hamiltonian Monte Carlo method for inference.

**Model comparison.**   WAIC has been utilized to determine the goodness-of-fit of each model. WAIC is defined as an estimate of out-of-sample deviance (the predictive accuracy for new samples) with an adjustment for in-sample deviance (overfitting to observed samples). WAIC values can be derived as follows [34, 43, 44]:

$$\text{WAIC} = -2(\text{lppd} - p_{\text{WAIC}}),$$
$$\text{lppd} = \sum_{n=1}^{N} \log\left(\frac{1}{S}\sum_{s=1}^{S}\text{Pr}(y_n|\Theta_s)\right),$$
$$p_{\text{WAIC}} = \sum_{n=1}^{N}\frac{1}{S-1}\sum_{s=1}^{S}\left(\log\text{Pr}(y_n|\Theta_s) - \frac{1}{S}\sum_{s'=1}^{S}\log\text{Pr}(y_n|\Theta_{s'})\right)^2, \quad (7)$$

where $n$, $N$, $s$, and $S$ represent the observation (data point), the total number of observations, the MCMC sample, and the total number of MCMC samples, respectively. $\text{Pr}(y_n|\Theta_s)$ is defined as the likelihood: the probability of $y$ in observation $n$ given the set of inferred parameters in sample $s$, $\Theta_s$. The lppd is the "log point wise predictive density" indicating predictive accuracy: the likelihood of each observation $n$ is averaged over samples, and the logarithm of the averaged likelihood is then summed up across the observations. The $p_{\text{WAIC}}$ is the "penalty term," representing the variance in the predictions: the variance in log likelihood over the samples is calculated for observation $n$, and each variance is then summed up across the observations. The smallest value of WAIC indicates the best model in terms of predicting the experimental data.

## Results

### Behavioral results

Fig 2 shows the distributions of participants' cooperation probabilities conditioned by the partner's previous decision. These probabilities are denoted by $p(C|C)$, and $p(C|D)$. Fig 3 shows the distributions of cooperation probabilities conditioned by both one's own and the partner's previous decisions, which are denoted by $p(C|CC)$, $p(C|DC)$, $p(C|CD)$, and $p(C|DD)$.

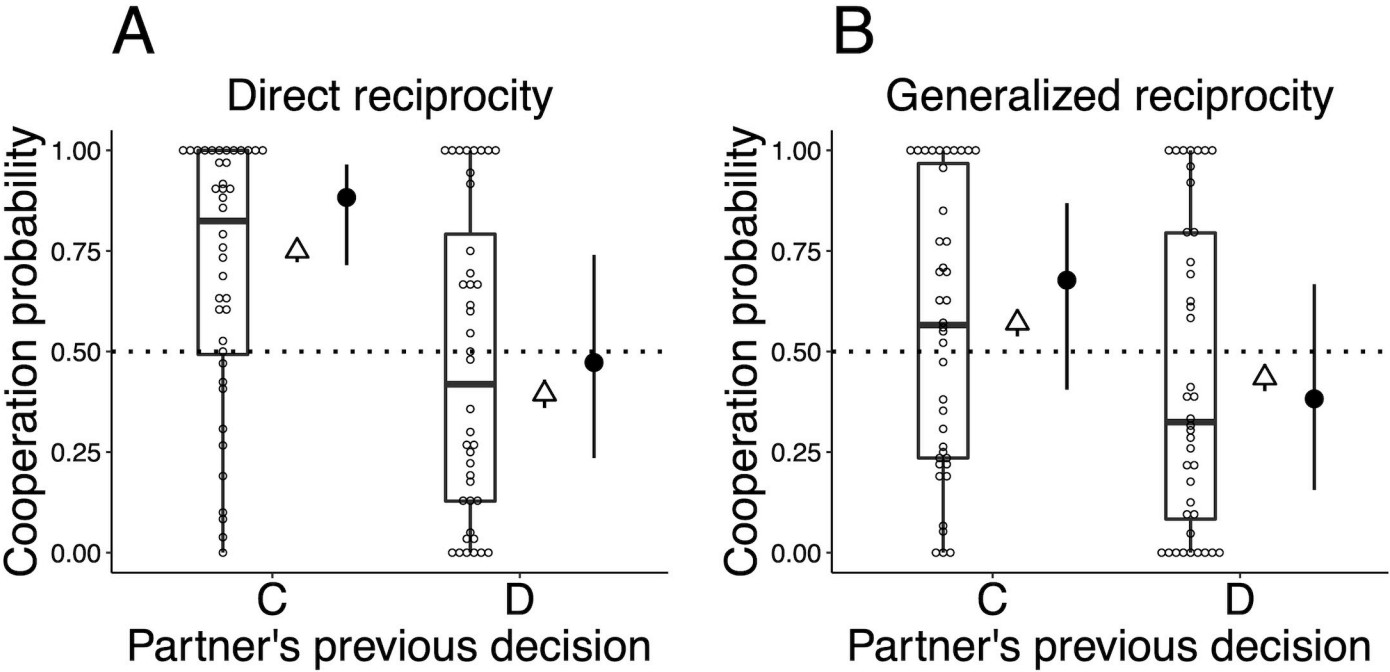

**Fig 2. Distributions of each cooperation probability calculated separately for the partner's previous decision.** (A) Direct reciprocity game. (B) Generalized reciprocity game. Boxplots indicate the participant's empirical cooperation probabilities: $p(C|C)$ and $p(C|D)$. The point on each boxplot, the box, the thick line in each box, and the whisker represent each participant, the interquartile range (IQR), the median, and the distances $1.5 \times$ IQR, respectively. The open triangles represent the overall fraction of cooperation averaged over all participants (the error bars represent 95% confidence intervals: $\pm 1.96 \times$ standard error). The filled circles and bars adjacent to the right-hand side of each boxplot indicate the predicted distributions of group-level cooperation probabilities inferred from the partner's action (PA) model, $\hat{p}(C|C)$ and $\hat{p}(C|D)$. Each filled circle and bar represent the median and the 95% compatibility interval of the predicted distribution, respectively. Each label on the horizontal axis indicates the partner's decision in the previous round: C, the partner cooperated; and D, he/she defected.

Figs 2 and 3 also present the fraction of cooperation averaged over all participants (the open triangles in Figs 2 and 3) and empirical cooperation probabilities calculated for each participant (the open circles in Figs 2 and 3). S2 Text details the methods used to calculate these cooperation probabilities. As Fig 2 indicates, the averaged fraction of cooperation after the partner has decided to cooperate, $p(C|C)$, in the direct reciprocity game is higher than the chance level (namely, 50%), whereas in the generalized reciprocity game it is near the chance level. Fig 3 shows that in both games the averaged fraction of cooperation after both the focal player and his/her partner have cooperated, $p(C|CC)$, is higher than in the other three cases.

## Model comparison

Table 1 presents the WAIC values for each model. The smaller the WAIC value of a model, the better is its prediction performance. For each model, Table 1 also reports $p_{WAIC}$, standard error (SE) of the WAIC, difference in the WAIC between each model and the best model (dWAIC), standard error of the dWAIC (dSE), and the weight of the dWAIC. The weight can be interpreted as the relative distances between the WAIC of the best model and that of the other considered models: the weight for a model $k$ ($w_k$) is calculated as follows [44]:

$$w_k = \frac{\exp(-0.5 \mathrm{dWAIC}_k)}{\sum_{j=1}^{J} \exp(-0.5 \mathrm{dWAIC}_j)}, \tag{8}$$

where $\mathrm{dWAIC}_k$ represents the dWAIC of model $k$.

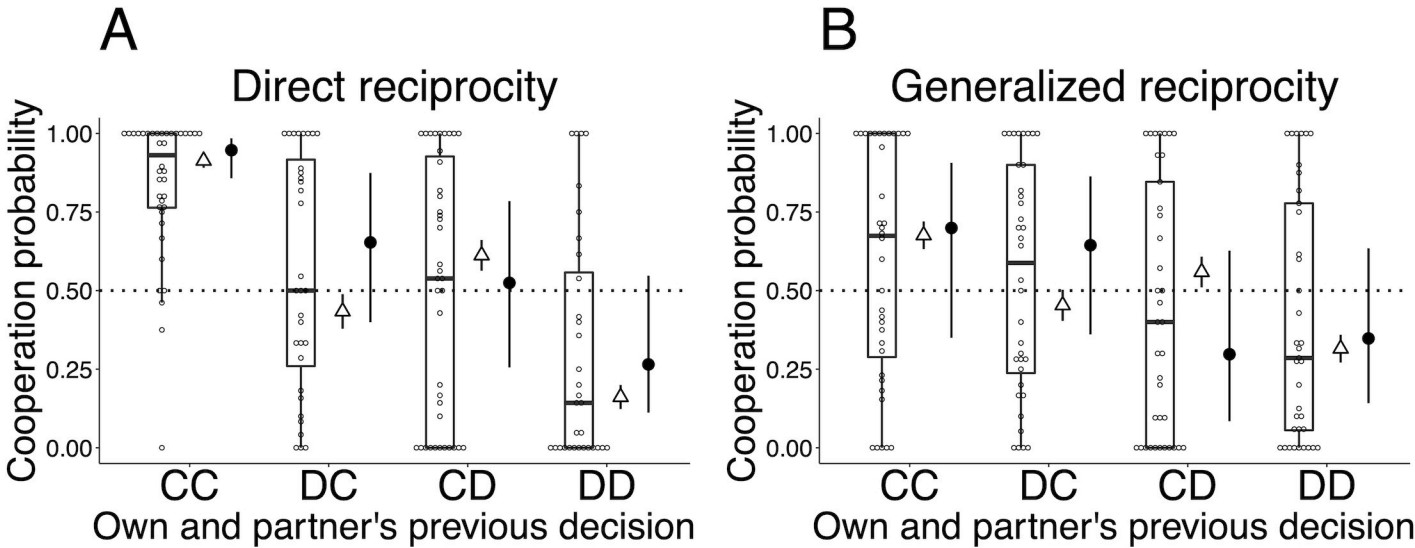

**Fig 3. Distributions of each cooperation probability calculated separately for the combination of one's own and the partner's previous decisions.** (A) Direct reciprocity game. (B) Generalized reciprocity game. Boxplots indicate the participant's empirical cooperation probabilities: $p(C|CC)$, $p(C|DC)$, $p(C|CD)$, and $p(C|DD)$. The point on each boxplot, the box, the thick line in each box, and the whisker represent each participant, the IQR, the median, and the distances $1.5 \times$ IQR, respectively. The open triangles represent the overall fraction of cooperation averaged over the participants (error bars represent 95% confidence intervals: $\pm 1.96 \times$ standard error). The filled circles and bars adjacent to the right-hand side of each boxplot indicate the predicted distributions of group-level cooperation probabilities inferred from the OPA model, $\hat{p}(C|CC)$, $\hat{p}(C|DC)$, $\hat{p}(C|CD)$, and $\hat{p}(C|DD)$. Each filled circles and bar represent the median and the 95% compatibility interval of the predicted distribution, respectively. Each label on the horizontal axis indicates the participant's and his/her partner's decision in the previous round: CC, both players had cooperated; DC, the participant defected while his/her partner cooperated; CD, the participant cooperated while his/her partner defected; and DD, both players defected.

Table 1 shows that the OPA model has the smallest WAIC value and the highest weight among all the considered models for both the direct and generalized reciprocity games. This indicates that the OPA model has the best performance in terms of predicting the experimental data, regardless of the game type. In both games, the second-best model was the PA model, and the third best was the OA model. The WAIC values of the null model is larger than those of the other three models. Note that in both games, the OPA model predicts the data better

**Table 1. WAIC values for each multilevel model.**

|  | WAIC | $p_{\text{WAIC}}$ | dWAIC | SE | dSE | weight |
|---|---|---|---|---|---|---|
| Direct reciprocity game |  |  |  |  |  |  |
| Own and partner's action (OPA) | 1113.02 | 76.17 | 0 | 45.33 | NA | 1 |
| Partner's action (PA) | 1190.76 | 55.06 | 77.74 | 44.23 | 22.01 | 0 |
| Own action (OA) | 1388.38 | 54.77 | 275.36 | 43.08 | 29.76 | 0 |
| Null | 1428.57 | 32.22 | 315.55 | 41.98 | 35.88 | 0 |
| Generalized reciprocity game |  |  |  |  |  |  |
| OPA | 1238.66 | 77.92 | 0 | 43.28 | NA | 1 |
| PA | 1312.85 | 55.32 | 74.18 | 41.68 | 19.34 | 0 |
| OA | 1734.43 | 58.44 | 495.76 | 35.73 | 35.50 | 0 |
| Null | 1790.07 | 33.55 | 551.41 | 33.15 | 38.70 | 0 |

dWAIC = difference between WAIC of each model and that of the best model (i.e., the OPA model); SE = standard error of each WAIC; dSE = The standard error of the dWAIC; weight = the weight of dWAIC.

than the PA model, even though the OPA model has more parameters and thus risks overfitting to the data.

## Group-level cooperation probabilities inferred from the PA and OPA model

To check whether each model can predict the group-level cooperation probabilities well, the predicted distributions of the cooperation probabilities are inferred for both the PA and the OPA model. The group-level probability of cooperation after the partner has cooperated, denoted by $\hat{p}(C|C)$, and that after the partner's defection, denoted by $\hat{p}(C|D)$, were inferred from the PA model. Similarly, the group-level cooperation probabilities, denoted by $\hat{p}(C|CC)$, $\hat{p}(C|DC)$, $\hat{p}(C|CD)$, and $\hat{p}(C|DD)$, respectively, conditioned according to the combinations of one's own and the partner's previous actions, were inferred from the OPA model (see S2 Text for details). The predicted distributions of group-level cooperation probabilities are also shown in Figs 2 and 3 (i.e., filled circles and bars).

As Fig 2 indicates, in both games, the distribution of $\hat{p}(C|C)$ and $\hat{p}(C|D)$ predicted by the PA model overlaps the empirical overall fractions of cooperation (the open triangles in the Fig). Therefore, the PA model predicts well the group-level cooperation probability conditioned by the partner's previous action. Fig 3 also indicates that, in both games, the distributions of $\hat{p}(C|CC)$, $\hat{p}(C|DC)$, $\hat{p}(C|CD)$, and $\hat{p}(C|DD)$ predicted by the OPA model overlap the empirical fraction of cooperation conditioned by one's own and the partner's previous actions.

## Difference between group-level cooperation probabilities

To compare the differences between cooperation probabilities, Fig 4 presents the difference of predicted distributions between the group-level inferred cooperation probabilities. Fig 4 also show the probabilities that each difference can be greater than 0 (namely, the shaded area of each distribution).

In the direct reciprocity game, $\hat{p}(C|CC)$ is greater than the other three probabilities, while $\hat{p}(C|DD)$ is lower than the other ones. Therefore, mutual cooperation enhanced the probability of cooperation in the subsequent decision more than in the other cases, whereas mutual defection suppressed it.

In contrast to the direct reciprocity game, a greater difference between $\hat{p}(C|CC)$ and $\hat{p}(C|DC)$ was not observed in the generalized reciprocity game. The patterns corresponding to the predicted distributions of $\hat{p}(C|CD) - \hat{p}(C|DD)$ in the generalized reciprocity game also differ from those in the direct one. These patterns of difference between the probabilities in the generalized reciprocity game suggest that mutual cooperation or defection did not have a great effect on enhancing or suppressing the subsequent cooperation probability in the generalized reciprocity game compared to the direct one. The cooperation probabilities after the partner has cooperated (namely, $\hat{p}(C|CC)$ and $\hat{p}(C|DC)$) are greater than the probabilities after the partner has defected (namely, $\hat{p}(C|CD)$ and $\hat{p}(C|DD)$), regardless of the participant's own previous action.

## Individual differences in behavioral patterns among participants

We compared inter-individual variations in behavioral patterns between the two games. The OPA model inferred varying effects for each participant, which determined the cooperation probabilities of each participant (namely, $\beta_{1,i}$, $\beta_{2,i}$, $\beta_{3,i}$, and $\beta_{4,i}$). The behavioral patterns of each participant are classified according to the posterior distributions of the individuals' parameter values. One could list a large quantity of patterns by considering the combination of

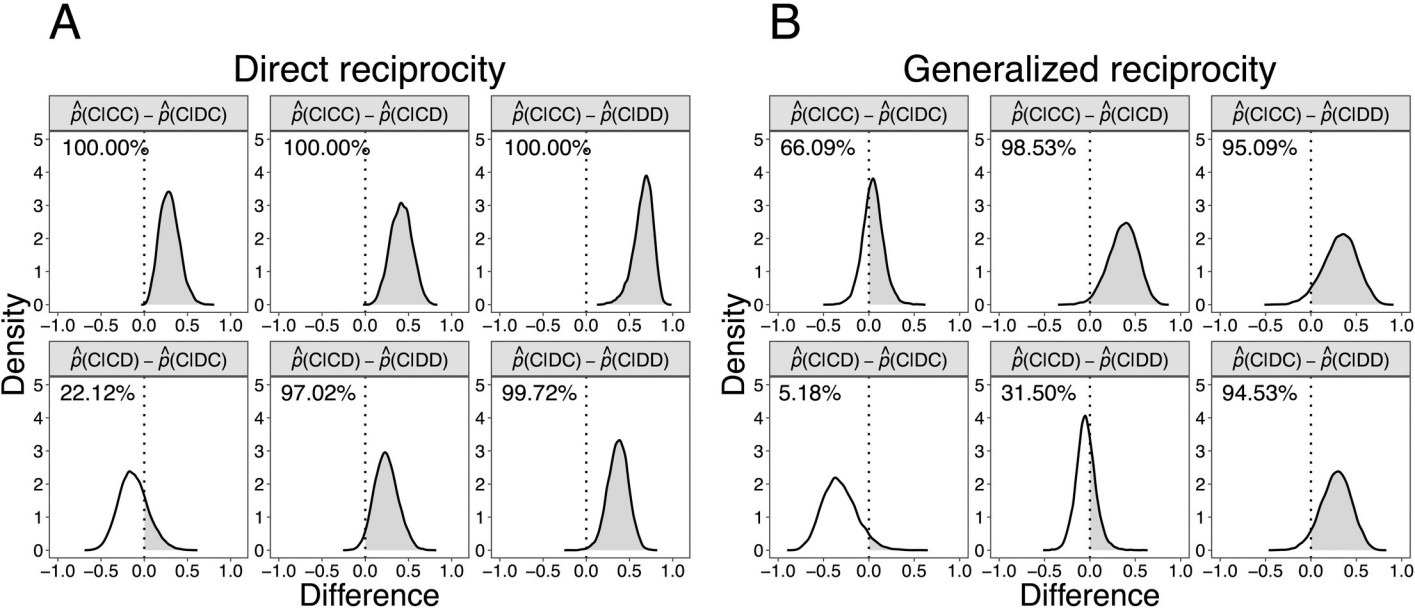

**Fig 4. The difference of predicted distributions between each cooperation probability, as inferred from the own and partner's action (OPA) model.** (A) Direct reciprocity game. (B) Generalized reciprocity game. The predicted distributions were estimated from 12,000 samples retrieved from Markov chain Monte Carlo (MCMC) simulations. The percentages shown in the upper left of each panel indicate the probability that the difference between the probabilities of cooperation is greater than 0 (i.e., the value of the percentage is equal to the shaded area of the distribution in each panel).

all four cooperation probabilities (namely, $\hat{p}(C|CC)_i$, $\hat{p}(C|DC)_i$, $\hat{p}(C|CD)_i$, and $\hat{p}(C|DD)_i$) or the differences between them. To simplify the categorization of behavioral patterns, we investigated how many participants are classified into TFT-like or WSLS-like strategies according to the predicted distributions of $\hat{p}(C|CC)_i$ and its difference from the other three probabilities.

Table 2 summarizes the rules used to classify the participants' behavioral types. First, we classified the participants either as those who tend to cooperate more after mutual cooperation (Types 1, 2, and 3) or as those who do not (Type 4), according to whether or not the 95% compatibility intervals for $\hat{p}(C|CC)_i$ are greater than 50%. Second, we classified those whose $\hat{p}(C|CC)_i$ is greater into one of three types, according to the 95% compatibility intervals for the difference between $\hat{p}(C|CC)_i$ and the other three probabilities: TFT-like (Type 1), WSLS-like (Type 2), or others (Type 3). Theoretically, WSLS should cooperate after mutual defection. However, most participants in fact cooperated less after mutual defection in our experiments, and several previous empirical studies have considered $p(C|DC)$ as difference between TFT-like and WSLS-like strategy [32, 33]. Therefore, we distinguish a TFT-like from a WSLS-like strategy according to whether or not $\hat{p}(C|CC)_i - \hat{p}(C|DC)_i$ is greater than 0. In S3 and S4 Figs,

**Table 2. Classification rules for behavioral types.**

|  | $\hat{\boldsymbol{p}}(C|CC)_i$ | $\hat{\boldsymbol{p}}(C|CC)_i - \hat{\boldsymbol{p}}(C|DC)_i$ | $\hat{\boldsymbol{p}}(C|CC)_i - \hat{\boldsymbol{p}}(C|CD)_i$ | $\hat{\boldsymbol{p}}(C|CC)_i - \hat{\boldsymbol{p}}(C|DD)_i$ |
|---|---|---|---|---|
| Type 1 (TFT-like) | >.50 | ≤ 0 | > 0 | > 0 |
| Type 2 (WSLS-like) | >.50 | > 0 | > 0 | - |
| Type 3 | >.50 | Other patterns except for above two | | |
| Type 4 | ≤.50 | Any patterns | | |

We assessed the behavioral types of each participant according to whether or not the 95% compatibility intervals of the predicted $\hat{p}(C|CC)$ distribution and its difference from other probabilities excludes 50% or 0.

**Table 3. Frequencies of each behavioral type in the direct and generalized reciprocity games.**

|  | Type 1 (TFT-like) | Type 2 (WSLS-like) | Type 3 | Type 4 |
|---|---|---|---|---|
| Direct reciprocity | 8 (.200) | 4 (.100) | 15 (.375) | 13 (.325) |
| Generalized reciprocity | 8 (.200) | 1 (.025) | 4 (.100) | 27 (.675) |

Numerical values in each parenthesis represent the proportions of each behavioral type in each game.

the predicted distributions of cooperation probabilities and differences between them are shown for each participant.

Table 3 shows the frequencies of each behavioral type in the two games. The distributions of each behavioral type differed significantly between the direct and generalized reciprocity games (Fisher's exact test: $p < .01$). The behavioral types for which $\hat{p}(\text{C}|\text{CC})_i$ is greater (namely, Types 1, 2, and 3) are observed more in the direct reciprocity game than in the generalized one. In the generalized reciprocity game, most participants are classified into Type 4, and it seems that the participants' behavioral patterns in that game vary more than in the direct one (S3 and S4 Figs). In both games, some proportions of TFT-like strategies are also observed. In the generalize reciprocity game, the WSLS-like and other behavioral patterns for which $\hat{p}(\text{C}|\text{CC})_i$ is greater than other three probabilities (namely, Type 3) are hardly observed, as compared to the direct reciprocity game.

## Comparison between the multilevel model and another method

As a supplementary analysis for comparing the multilevel model to another method for estimating inter-individual differences, we fitted a non-multilevel OPA model to the data: a "non-pooling" OPA model. The "non-pooling" OPA model separately inferred each parameter for each participant ($\beta_{1,i}, \beta_{2,i}, \beta_{3,i}$, and $\beta_{4,i}$) but did not assume that the parameters for each participant obeyed the normal distribution: the model independently estimated the individual-level parameters. For comparison with the multilevel and the non-pooling OPA model, we also fitted a "pooling OPA" model, which assumed that each parameter value ($\beta_1, \beta_2, \beta_3$, and $\beta_4$) was constant for all participants; i.e., the model ignored inter-individual differences.

As shown in S3 Table, in both the direct and generalized reciprocity games, the WAIC value of the multilevel OPA model was the smallest compared to the other two non-multilevel OPA models: the multilevel OPA model had greater predictive accuracy than both the non-pooling and the pooling OPA models. S5 Fig illustrates that the participants' parameters inferred by the non-pooling OPA model deviated from the group-level parameters inferred from the multilevel or the pooling OPA models (i.e., overfitting to the data occurred), and the errors seemed to be high. In contrast, the parameters for each participant inferred by the multilevel OPA model were near the group-level parameters inferred by the multilevel or the pooling OPA models (i.e., shrinkage was observed).

## Discussion

The purpose of the current study was to investigate a model that predicts human behavior in both the direct and generalized reciprocity situations. The results of a model comparison revealed that for both the direct and generalized reciprocity games, the model that takes into consideration both one's own and the partner's behaviors predicts the experimental data better than the model that uses only the partner's behavior as reference. The distributions of participants' behavioral types in each game also suggest that there are various individual strategies

and that the OPA model predicts such variations in behavioral types well despite its overfitting risk.

However, the results of the analysis also suggest that people adopt different strategies depending on the type of interaction. In the direct reciprocity situation, the participants generally cooperated or defected more after both the players and their partners had either cooperated or defected, respectively. On the other hand, the average behavioral tendency in the generalized reciprocity game differed from that in the direct reciprocity game. In the generalized reciprocity game, the WAIC value of the OPA model was the smallest among the all models, and various behavioral types that depend on both one's own and the partner's actions were observed. Nevertheless, differences between probabilities and a small number of behavioral types with greater $\hat{p}(\text{C|CC})$ suggest that the effect of one's own and one's partner's previous actions on the subsequent cooperation is weak in the generalized reciprocity situation.

The group-level behavioral patterns observed in the direct reciprocity game differed from both complete TFT and WSLS. Even though the partner had cooperated in the previous round, our participants cooperated more often after they had mutually cooperated than after they had defected. Classification of the participants' behavioral type also suggests that many behavioral types for which $\hat{p}(\text{C|CC})_i$ is greater (namely, Types 1, 2, and 3) were observed in the direct reciprocity game. Although WSLS was able to predict that the players would repeat the previous behavior yielding larger earnings, most of the participants did not tend to exploit cooperators and did not shift their behavior after mutual defection. In the field of social psychology, it has been argued traditionally that the motivation for cooperation in PD situations is grounded on both players' preference for cooperation and the expectation that his/her partner would cooperate [45]. A series of experiments have indicated that there exists a correlation between cooperation in social dilemmas and expectations about the opponent's cooperation [36, 37]. These arguments suggest that people's motivation behind cooperation would be based on preference for mutual cooperation rather than reaction to higher payoffs. The findings that mutual cooperation enhanced the subsequent cooperation in the direct reciprocity game would be consistent with these arguments.

However, the achievement of mutual cooperation would not be expected in the generalized reciprocity game because of a lack of control of the neighbor's behavior. In contrast to the direct reciprocity game, it would be difficult for costly cooperation in generalized reciprocity to be rewarded. Therefore, although the OPA model fit to the data better than other models, a weaker effect of mutual cooperation on subsequent cooperation might be observed in the generalized reciprocity game. Similar to previous empirical studies [10, 11], the experimental situation of the current research with a small population and a short cycle of decision chains would relatively induce expectation for return of cooperation, even in the generalized reciprocity case. It has been suggested both theoretically [19, 20] and empirically [17] that cooperation based on generalized reciprocity may be relatively fragile in other conditions, such as a large group size. It is possible that other behavioral models rather than the OPA model may also be suitable for generalized reciprocity in different experimental conditions in which the return of cooperation could be expected to be less. It is thus necessary to investigate whether or not behavioral patterns we observed in the generalized reciprocity case are consistent across different conditions that correspond to a real human society, such as a large group.

An alternative method for estimating individual behavioral patterns is to fit the models to each participant separately using the maximum likelihood method. However, the results from such an analysis can be uncertain when the data sample is limited or there exists an imbalance of cases among individuals. In behavioral experiments using social interactions, even if we increase the number of rounds of decisions, an imbalance of decision cases among the

participants may occur. For example, the number of times each participant receives help may differ among participant. In fact, in our experimental data analysis that estimated parameter values independently by each participant (i.e., fitting the non-pooling OPA to the data) seemed to produce uncertain results and overfit to the data. In such a case, a multilevel model with Bayesian inference can provide reliable estimates even though there may be an imbalance in the samples, by inferring the intervals of parameters and shrinkage to the group-level means [44, 46]. As suggested by our analysis comparing the multilevel and non-multilevel models, multilevel modeling can thus be a useful tool for modeling human behavior in social interactions.

However, in this study, the question of the individual's consistency of behavior across games still remains unclear, as the participants played either of two games. Several previous studies in which the same participants played various experimental games indicated positive correlations of cooperative behavior between games and argued for the existence of a domain-general pro-sociality [36, 47]. The current study conducted dyadic interactions as a basis for human interactions, but it should be extended to other situations, such as social dilemmas [24–30]. Further investigation is required to confirm the individual consistency of strategies across different domains.

Evolutionary game theory has provided a description of the evolution of human cooperation. The theory should be tested as to whether people adopt the strategies assumed in the theoretical models, and various empirical studies using laboratory experiments have examined this issue [4, 5]. The combination of laboratory experimental methods and analytical approaches to human behavioral data would allow the derivation of fruitful implications concerning both theoretical and empirical investigations of human cooperation.

## Supporting information

**S1 Fig. Samples of the decision screens displayed to the participants.**
(PDF)

**S2 Fig. Example of simultaneously running three independent chains of decisions in the generalized reciprocity game.**
(PDF)

**S3 Fig. Individual parameter values inferred separately for each participant in the direct reciprocity game.**
(PDF)

**S4 Fig. Individual parameter values inferred separately for each participant in the generalized reciprocity game.**
(PDF)

**S5 Fig. Individual parameter values inferred by the multilevel, non-pooling, and pooling own and partner's action (OPA) model.**
(PDF)

**S1 Table. Posterior distributions of the parameters for each model in the direct reciprocity game.**
(PDF)

**S2 Table. Posterior distributions of the parameters for each model in the generalized reciprocity game.**
(PDF)

**S3 Table. WAIC values for the multilevel, non-pooling, and pooling own and partner's action (OPA) model.**
(PDF)

**S1 Text. Parameter inference for the multilevel models.**
(PDF)

**S2 Text. Method for calculating the cooperation probabilities.**
(PDF)

## Acknowledgments

I acknowledge Masanori Takezawa for his helpful comments on the manuscript and the colleagues at the Department of Psychology, Teikyo University, for their cooperation in recruiting participants. I would like to thank Enago (www.enago.jp) for the English language review.

## Author Contributions

**Conceptualization:** Yutaka Horita.

**Data curation:** Yutaka Horita.

**Formal analysis:** Yutaka Horita.

**Funding acquisition:** Yutaka Horita.

**Investigation:** Yutaka Horita.

**Methodology:** Yutaka Horita.

**Project administration:** Yutaka Horita.

**Resources:** Yutaka Horita.

**Software:** Yutaka Horita.

**Visualization:** Yutaka Horita.

**Writing – original draft:** Yutaka Horita.

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
