## [Decision Letter · Decision Letter 0]

2 Oct 2020

PONE-D-20-26747

Mutual cooperation and defection affect subsequent cooperation in direct reciprocity: Behavioral experiments and analysis using multilevel models.

PLOS ONE

Dear Dr. Horita,

Thank you for submitting your manuscript to PLOS ONE. After careful consideration, we feel that it has merit but does not fully meet PLOS ONE’s publication criteria as it currently stands. Therefore, we invite you to submit a revised version of the manuscript that addresses the points raised during the review process.

We look forward to receiving your revised manuscript.

Kind regards,

Valerio Capraro

Academic Editor

PLOS ONE

Additional Editor Comments:

I have now received one review from one expert in the field. The reviewer finds the idea interesting but suggests several improvements, especially in the writing and in the analysis, which should be addressed in a major revision. I was unable to find another reviewer, but this review is very detailed - certainly more than the average review - therefore I feel confident making a decision based on this review and my own reading of the paper. Therefore, I would like to invite you to revise your work following the reviewer's comments.

I am looking forward for the revision.

Journal Requirements:

2. Please change "female” or "male" to "woman” or "man" as appropriate, when used as a noun.

Reviewers' comments:

Reviewer's Responses to Questions

**Comments to the Author**

1. Is the manuscript technically sound, and do the data support the conclusions?

Reviewer #1: Partly

2. Has the statistical analysis been performed appropriately and rigorously? 

Reviewer #1: No

3. Have the authors made all data underlying the findings in their manuscript fully available?

Reviewer #1: Yes

4. Is the manuscript presented in an intelligible fashion and written in standard English?

Reviewer #1: Yes

5. Review Comments to the Author

Reviewer #1: This article studies two multilevel models and their effectiveness in predicting the cooperation in direct and generalized reciprocity from two behavioral experiments. The results suggest that the model that takes into account both one's own previous action and one's partner's previous action predicts better, and observes several subtypes of behavioral strategies including tit-for-tat and win-stay-lose-shift.

The overall motivation of the paper is quite interesting. The writing would benefit from additional clarifications on technical details. Other than the writing revisions, I believe this would require additional analyses to support several statements made in this article. The points of suggestion and concern is as follows:

Line 176 Fig 1. The caption should be self-contained without referencing the main text. It is not informative enough. Please elaborate.

Line 284. "v denotes the cooperative tendency". Please include more details. Does v range from 0 to 1, and refer to the probability to cooperate?

Line 292. "mean 0 and standard deviation 10" should be "mean of 0 and standard deviation of 10"

Line 315. "where lppd and pWAIC mean the log point-wise posterior predictive density and the effective number of parameters". Please provide more details for lppd and pWAIC. It is hard for the readers to decode the results in Table 1 and how the authors compute this two values exactly without necessary details and references.

Line 371. Fig 3 Caption: "Filled symbols and bars adjacent to the right side of each boxplot indicate the predicted distributions of group-level cooperation probabilities inferred from the OPA model". Not exactly. The grey filled symbols actually corresponds to the PA model. Please clarify it with more details to avoid the confusion.

Line 409/414. The presentation of Fig 2 and Fig 3 (a main discovery) is lack of proper statistical tests to support the argument. "overlaps the empirical overall fractions of cooperation" is a vague and loose evaluation. The readers cannot conclude whether the prediction from OPA and PA are significantly different from the behavioral data or not.

Line 403/457/458 and more. The usage of square bracket and parentheses are inconsistent throughout the entire paper: e.g. use "(namely, p(C|DD))" instead of "[namely, p(C|DD)]"

Line 433/438. Incorrect statement. "predicted distributions of the difference" should be "the difference of predicted distributions". The model is not predicting the difference.

Line 455 and Fig 4. In the case of p̂(C|CD)– p̂(C|DD), the generalized reciprocity game is very different from that of the direct one, which likely suggest the candidate multi-level models are predicting generalized reprocity poorly. As the authors also pointed out in Line 557, the lack of mutual reciprocity would make mutual cooperation not expected in the generalized condition. Thus, the choice of using OPA and PA seems a little far-fetched to apply to the generalized condition in the first place. Please clarify the rationale.

Table 3. Incorrect rounding. In "Direct reciprocity", number 13 should correspond to (.33) instead of (.32). In "Generalized reciprocity" 1 should correspond to (.03) instead of (.02).

Line 552 "If humans generally did not react to the objective value of earnings but subjectively weighted higher the value corresponding to mutual cooperation, the behavioral patterns observed in the direct reciprocity game would be consistent with the argument of human pro-sociality." This sentence is a little hard to comprehend. Please clarify.

The authors argues that a popular alternative method, maximum likelihood method, "can be uncertain when the data sample is limited or there exists an imbalance of cases among individuals". I find the conclusion of not using it unconvincing. Please consider include maximum likelihood method as a comparison to support the previous statement.

One missing modeling component, is the lack of a model that only takes into account of one's own previous action, in another word, Own Action (OA) model. Please consider including this condition. The results in Fig 3 suggest that OPA's predicted cooperation probability is quite different from that from PA. One likely factor could be simply one's own previous action. Please consider including it.

The authors also didn't convince me that the direct and generalized reciprocity game settings are comparable in the first place. The direct reciprocity game, the cooperation and defects have different stages of risks and rewards, as well as a mutual cooperation component. There is a "dilemma" component involved. The generalized reciprocity game, however, at least in this specific experimental setting proposed by the authors, involves monetary gains that have no serious consequences of defecting. The contradicting results of p̂(C|CD)– p̂(C|DD) in Fig 4 also supports my concern that the generalized reciprocity case is an entirely different case that might be insuitable to be compared with the direct version.

Title inconsistency: The main title is "Mutual cooperation and defection affect subsequent cooperation in direct reciprocity". The paper, however, devoted around half of the space investigating also the generalized reciprocity.

6. PLOS authors have the option to publish the peer review history of their article (what does this mean?). If published, this will include your full peer review and any attached files.

Reviewer #1: No

---

## [Author Response · Author response to Decision Letter 0]

21 Oct 2020

Response to Reviewer #1

We are grateful to the reviewer for carefully reading our manuscript and providing valuable comments. We have revised the issues raised by the reviewer. 

>Reviewer #1: This article studies two multilevel models and their effectiveness in predicting the cooperation in direct and generalized reciprocity from two behavioral experiments. The results suggest that the model that takes into account both one's own previous action and one's partner's previous action predicts better, and observes several subtypes of behavioral strategies including tit-for-tat and win-stay-lose-shift.

>The overall motivation of the paper is quite interesting. The writing would benefit from additional clarifications on technical details. Other than the writing revisions, I believe this would require additional analyses to support several statements made in this article. The points of suggestion and concern is as follows:

Thank you very much for your careful reading. We hope that we have successfully addressed the issues raised by the reviewer. Please find our replies to your comments below. Please see the revised manuscript with tracking changes.

>Line 176 Fig 1. The caption should be self-contained without referencing the main text. It is not informative enough. Please elaborate.

We added brief descriptions of each game to the Fig 1 caption. Please see the lines 196–203.

>Line 284. "v denotes the cooperative tendency". Please include more details. Does v range from 0 to 1, and refer to the probability to cooperate?

For all the models and all the parameters, there are no limitations for the parameter value ranges. We described the parameter range in the description of each model. Inferred parameter values with a free range were transformed into probabilities ranging from 0 to 1 using the inverse logit function (i.e., p = exp(x)/(1 + exp(x))). In the revised manuscript, we added this explanation in lines 310–314. 

>Line 292. "mean 0 and standard deviation 10" should be "mean of 0 and standard deviation of 10"

Thank you for your careful reading. We have corrected this in the manuscript in lines 291, 294, and 324–325.

>Line 315. "where lppd and pWAIC mean the log point-wise posterior predictive density and the effective number of parameters". Please provide more details for lppd and pWAIC. It is hard for the readers to decode the results in Table 1 and how the authors compute this two values exactly without necessary details and references.

We described how to calculate lppd and pWAIC in Equation (7) and the main text. We cited the following references for understanding these values. 

Gelman A, Hwang J, Vehtari A. Understanding predictive information criteria for Bayesian models. Stat Comput. 2014; 24: 997–1016. doi: 10.1007/s11222- 013-9416-2

McElreath R. Statistical Rethinking: A Bayesian Course with Examples in R and Stan (Second Edition). Boca Raton, FL: CRC Press; 2015.

Please see lines 377–391.

>Line 371. Fig 3 Caption: "Filled symbols and bars adjacent to the right side of each boxplot indicate the predicted distributions of group-level cooperation probabilities inferred from the OPA model". Not exactly. The grey filled symbols actually corresponds to the PA model. Please clarify it with more details to avoid the confusion.

Thank you for pointing out our mistake. However, as described in the next response, we removed the predictive distributions of the PA model from Fig. 3. Please see below.

>Line 409/414. The presentation of Fig 2 and Fig 3 (a main discovery) is lack of proper statistical tests to support the argument. "overlaps the empirical overall fractions of cooperation" is a vague and loose evaluation. The readers cannot conclude whether the prediction from OPA and PA are significantly different from the behavioral data or not.

In Table 1, we added information for the difference between the WAIC of each model and that of the best model. Similar reports have been shown in the following works, which conducted model comparisons using Bayesian inference:

Brand CO, Mesoudi A. 2019. Prestige and dominance-based hierarchies exist in naturally occurring human groups, but are unrelated to task-specific knowledge. R. Soc. Open Sci. 6: 181621. http://dx.doi.org/10.1098/rsos.181621

Mesoudi A (2020). Cultural evolution of football tactics: strategic social learning in managers’ choice of formation. Evolutionary Human Sciences 2, e25, 1–14.

https://doi.org/10.1017/ehs.2020.27

In contrast to null hypothesis significance testing, in the context of Bayesian analysis, there is still no consensus on how much difference in the WAIC values between models can be considered as a “significant” difference (McElreath, 2015). However, as shown by the weight of difference in the WAIC among all the considered models, it would be obvious that for both games, the OPA model had relatively better predictive performance than the other three candidate models. 

In addition, as the weights of the WAIC suggested, it would be difficult to argue that the PA model still had sufficient predictive accuracy for the data compared to the OPA model in the generalized reciprocity game. As you pointed out, comparing predicted distributions between p(C|CC) and p(C|C), which were inferred from different models, may be too loose an evaluation and not compatible because the cases were also different when we estimated p(C|CC) and p(C|C). Therefore, as a conservative view, we modified the descriptions of the predictive accuracy of the PA model in the generalized reciprocity game and removed the predicted distributions inferred from the PA model in Fig 3. 

>Line 403/457/458 and more. The usage of square bracket and parentheses are inconsistent throughout the entire paper: e.g. use "(namely, p(C|DD))" instead of "[namely, p(C|DD)]"

We have removed all the square brackets throughout the manuscript, except for the citations. 

>Line 433/438. Incorrect statement. "predicted distributions of the difference" should be "the difference of predicted distributions". The model is not predicting the difference.

Thank you for pointing this out; we have corrected it. Please see lines 516 and 522. We also corrected the captions in the Supplementary S3 and S4 Figs. 

>Line 455 and Fig 4. In the case of p̂(C|CD)– p̂(C|DD), the generalized reciprocity game is very different from that of the direct one, which likely suggest the candidate multi-level models are predicting generalized reprocity poorly. As the authors also pointed out in Line 557, the lack of mutual reciprocity would make mutual cooperation not expected in the generalized condition. Thus, the choice of using OPA and PA seems a little far-fetched to apply to the generalized condition in the first place. Please clarify the rationale.

First, we added the interpretation of what p̂(C|CD)– p̂(C|DD) and p̂(C|DC)– p̂(C|CC) meant in lines 540–544. These tendencies of differences suggested that the effects of mutual cooperation and defection on subsequent cooperation would be weak in the generalized reciprocity situations. 

In the introduction, I have added a paragraph to explain the prediction of how the effect of mutual cooperation on increase in cooperation differed between the two games and the purpose of fitting the OPA model. Please see lines 149–162.

As you pointed out, the behavioral patterns in the generalized reciprocity game would differ from those in the direct reciprocity game. For direct reciprocity, cooperation would be immediately rewarded; therefore, players could increase their payoff by achieving mutual cooperation. However, such an immediate return of cooperation would not be expected in the generalized reciprocity. Thus, how important players considered achieving mutual cooperation as a goal would differ between the two games. However, several theoretical models considering the WSLS have been proposed for explaining generalized reciprocity. Therefore, we considered the OPA model as a candidate model in the generalized reciprocity game for comparison with the direct reciprocity situation. 

>Table 3. Incorrect rounding. In "Direct reciprocity", number 13 should correspond to (.33) instead of (.32). In "Generalized reciprocity" 1 should correspond to (.03) instead of (.02).

Thank you for your careful reading. However, when we rounded these numbers, the summed values of the proportions became 1.01. Therefore, we showed the proportions with three digits (e.g., 0.325). 

>Line 552 "If humans generally did not react to the objective value of earnings but subjectively weighted higher the value corresponding to mutual cooperation, the behavioral patterns observed in the direct reciprocity game would be consistent with the argument of human pro-sociality." This sentence is a little hard to comprehend. Please clarify.

Thank you for pointing this out. We have simplified this sentence. Please see lines 666–669.

>The authors argues that a popular alternative method, maximum likelihood method, "can be uncertain when the data sample is limited or there exists an imbalance of cases among individuals". I find the conclusion of not using it unconvincing. Please consider include maximum likelihood method as a comparison to support the previous statement.

Thank you for proposing this additional analysis. At first, we tried to estimate the parameters separately for each participant using the maximum likelihood estimation (i.e., point estimation of the parameter), but we quit because varying results were observed depending on the setting of the initial values to explore the parameter values. We guessed that this was due to the small number of decisions for each participant. 

As another alternative method for estimating the individual-level parameter, we estimated the posterior distributions of the parameters separately for each participant using Bayesian inference. We fitted three OPA models to the data: (1) the multilevel OPA model, which had already been reported in the main text; (2) the pooling OPA model, which assumed that the parameter values were constant across all participants; and (3) the non-polling OPA model, which considered inter-individual differences of the parameters but did not assume that the parameters for each individual obeyed the normal distribution. The results revealed that the multilevel OPA model still fit to the data better than the other two OPA models for both games: the WAIC of the multilevel model was smallest among these models. The estimated parameters from the non-pooling OPA model deviated from the group-level parameters inferred from the multilevel and the pooling OPA model (i.e., overfitting to the data of each individual), and the errors seemed to be high; whereas the multilevel OPA model estimated parameters were near the group-level parameters (i.e., shrinkage was observed). Thus, in our data, the multilevel method might be more appropriate for predicting the data than the other methods for estimating inter-individual differences by adjusting the uncertainty and overfitting to data.

We added additional analysis in the results section (lines 603–624) and mentioned these results in the discussion (lines 703–706 and 708–710). We have added the codes for the above supplementary analysis to the Open Science Framework repository.

>One missing modeling component, is the lack of a model that only takes into account of one's own previous action, in another word, Own Action (OA) model. Please consider including this condition. The results in Fig 3 suggest that OPA's predicted cooperation probability is quite different from that from PA. One likely factor could be simply one's own previous action. Please consider including it.

We added the result using the own action (OA) model, which considered only one’s own previous action for predicting the probability of cooperation. Please see lines 340–347 for the description of the model. In lines 465–466 and Table 1, we added the result of the inference of the OA model. The conclusion was not changed: the OPA model was still the best model for both the games, even when we included the OA model for the model comparison. 

The description of the OA model followed the OPA model. Because we could not hypothesize that only one’s own previous action affected subsequent cooperation by referring to previous studies, we used the OA model as a comparable model for the two main candidate models (i.e., the PA and the OPA model). 

>The authors also didn't convince me that the direct and generalized reciprocity game settings are comparable in the first place. The direct reciprocity game, the cooperation and defects have different stages of risks and rewards, as well as a mutual cooperation component. There is a "dilemma" component involved. The generalized reciprocity game, however, at least in this specific experimental setting proposed by the authors, involves monetary gains that have no serious consequences of defecting. The contradicting results of p̂(C|CD)– p̂(C|DD) in Fig 4 also supports my concern that the generalized reciprocity case is an entirely different case that might be insuitable to be compared with the direct version.

As mentioned in the previous response, in the introduction section, we described how the mutual cooperation would have different meanings in the direct and generalized reciprocity games (lines 149–162).

As a result of our model comparison, the OPA model was the best model, even for the generalized reciprocity game, but the effect of mutual cooperation on the subsequent cooperation seemed to be weaker in the generalized reciprocity game than in the direct one. In the discussion section, we speculated that the findings in the generalized reciprocity game would be associated with the difficulty in return of cooperation. Please see lines 675–695.

>Title inconsistency: The main title is "Mutual cooperation and defection affect subsequent cooperation in direct reciprocity". The paper, however, devoted around half of the space investigating also the generalized reciprocity.

We have changed the title. In the new title, we included both “direct” and “generalized reciprocity”.

In addition to the points described above; we have modified the following minor issues:

- I have changed “model selection” to “model comparison” for consistency with the term used in literatures (e.g., Mesoudi et al., 2020; Brand & Mesoudi, 2019; McElreath, 2015). 

- I have modified the parameter labels for each model according to the order the models are introduced.

Furthermore, some grammatical modifications were added by the English language review.

Finally, I would like to thank the reviewer for their comments and suggestions. I hope that the manuscript is now improved.

---

## [Decision Letter · Decision Letter 1]

6 Nov 2020

Greater effects of mutual cooperation and defection on subsequent cooperation in direct reciprocity games than generalized reciprocity games: Behavioral experiments and analysis using multilevel models.

PONE-D-20-26747R1

Dear Dr. Horita,

We’re pleased to inform you that your manuscript has been judged scientifically suitable for publication and will be formally accepted for publication once it meets all outstanding technical requirements.

Kind regards,

Valerio Capraro

Academic Editor

PLOS ONE

Additional Editor Comments (optional):

Reviewers' comments:

Reviewer's Responses to Questions

**Comments to the Author**

1. If the authors have adequately addressed your comments raised in a previous round of review and you feel that this manuscript is now acceptable for publication, you may indicate that here to bypass the “Comments to the Author” section, enter your conflict of interest statement in the “Confidential to Editor” section, and submit your "Accept" recommendation.

Reviewer #1: All comments have been addressed

2. Is the manuscript technically sound, and do the data support the conclusions?

Reviewer #1: (No Response)

3. Has the statistical analysis been performed appropriately and rigorously? 

Reviewer #1: (No Response)

4. Have the authors made all data underlying the findings in their manuscript fully available?

Reviewer #1: (No Response)

5. Is the manuscript presented in an intelligible fashion and written in standard English?

Reviewer #1: (No Response)

6. Review Comments to the Author

Reviewer #1: Thank you for revising the manuscript. Your responses and revisions have addressed all my previous concerns. Therefore, I recommend it to be accepted.

7. PLOS authors have the option to publish the peer review history of their article (what does this mean?). If published, this will include your full peer review and any attached files.

Reviewer #1: **Yes: **Baihan Lin

---

## [Editor Report · Acceptance letter]

10 Nov 2020

PONE-D-20-26747R1 

Greater effects of mutual cooperation and defection on subsequent cooperation in direct reciprocity games than generalized reciprocity games: Behavioral experiments and analysis using multilevel models. 

Dear Dr. Horita:

I'm pleased to inform you that your manuscript has been deemed suitable for publication in PLOS ONE. Congratulations! Your manuscript is now with our production department. 

Kind regards, 

on behalf of

Dr. Valerio Capraro 

Academic Editor

PLOS ONE